# Is There Such a Thing as “Anti-Nutrients”? A Narrative Review of Perceived Problematic Plant Compounds

**DOI:** 10.3390/nu12102929

**Published:** 2020-09-24

**Authors:** Weston Petroski, Deanna M. Minich

**Affiliations:** Human Nutrition and Functional Medicine Graduate Program, University of Western States, 2900 NE 132nd Ave, Portland, OR 97230, USA; wpetroski@students.uws.edu

**Keywords:** anti-nutrient, goitrogens, oxalates, phytates, phytoestrogens, plant-based diet, phytonutrients, tannins

## Abstract

Plant-based diets are associated with reduced risk of lifestyle-induced chronic diseases. The thousands of phytochemicals they contain are implicated in cellular-based mechanisms to promote antioxidant defense and reduce inflammation. While recommendations encourage the intake of fruits and vegetables, most people fall short of their target daily intake. Despite the need to increase plant-food consumption, there have been some concerns raised about whether they are beneficial because of the various ‘anti-nutrient’ compounds they contain. Some of these anti-nutrients that have been called into question included lectins, oxalates, goitrogens, phytoestrogens, phytates, and tannins. As a result, there may be select individuals with specific health conditions who elect to decrease their plant food intake despite potential benefits. The purpose of this narrative review is to examine the science of these ‘anti-nutrients’ and weigh the evidence of whether these compounds pose an actual health threat.

## 1. Introduction

Longstanding evidence suggests that consuming a diet rich in plant-based foods plays a significant role in prevention and reduction of chronic diseases, such as cardiovascular disease, cancer, stroke, dementia, diabetes, cataracts, and others [1,2]. Well-researched dietary patterns including the Mediterranean, Dietary Approaches to Stop Hypertension (DASH), vegan and vegetarian, as well as the hunter-gatherer (Paleolithic) diet, all provide ample amounts of whole foods in some capacity, including fruits, vegetables, nuts, legumes, and/or whole grains. Though specific aspects of these eating patterns may differ, they all encourage a variety of nutrient-dense, unprocessed plant foods, and reduced consumption of processed grains, added sugars and salt [3,4,5,6,7].

Similarly, the 2015–2020 Dietary Guidelines for Americans recommends eating a variety of nutrient-dense foods, particularly dark green vegetables, red and orange vegetables, and legumes [8]. Despite continuing educational efforts by the USDA, total fruit and vegetable intake remains exceedingly low, with less than 10% of Americans meeting recommended vegetable and fruit guidelines of 2.5 servings/day and two servings/day, respectively [9]. On the other hand, daily intakes of energy-dense refined grains are significantly higher than recommended values. Consuming a wide spectrum of plant foods ensures that individuals meet nutritional needs while staying within suggested energy requirements.

Plant-based foods, beyond micro- and macronutrients, contain significant concentrations of bioactive plant compounds. Research demonstrates that the reduction in chronic disease risk may be attributed to the synergistic effects of these anti-inflammatory phytochemicals, including an endless array of polyphenols, alkaloids, carotenoids, organosulfur compounds, terpenoids, and phytosterols [1,2]. Due to the diverse and complex interactions of vitamins, minerals and phytochemicals in a single food, health effects of a whole food, or combination of foods, will likely be significantly different than that of isolated compounds [10]. To complicate research even further, interaction of phytochemicals and microbiota within the intestinal environment could alter both bioavailability and biological effects [10,11]. For these reasons, elucidating the physiological effects of individual plant components obtained through dietary sources, composed of thousands of different compounds, is an implausible task.

More recently, various research has questioned the healthfulness of plant-foods because of the presence of certain compounds, termed ‘anti-nutrients’. These purported antinutrients, which include lectins, oxalates, phytates, phytoestrogens, and tannins, are thought to restrict bioavailability of key nutrients, while other studies conclude they may have health promoting effects [12,13] (Table 1). The purpose of this narrative review article is to provide an objective, scientific literature review of antinutrient compounds to assess whether they impose any significant health risk, and, further, whether they incur clinical implications.

## 2. Lectins

### 2.1. Definition

Lectins, or hemagglutinins, are a diverse family of carbohydrate-binding proteins found in almost all organisms, including plants, animals, and microorganisms [14]. These proteins/glycoproteins possess the unique capability to reversibly bind to specific carbohydrate moieties on cells, resulting in erythrocyte agglutination. The carbohydrate specificity of lectins allows them to function in cell recognition, tissue development, host defense and tumor metastasis in both plants and animals [15,16]. Over 500 lectins have been isolated and identified from plants, who produce them primarily as defense mechanisms against insects, molds, fungi and diseases [14].

### 2.2. Background

Plant lectins are widely distributed throughout the plant kingdom, available from many dietary sources including legumes, seeds, nuts, fruits, and vegetables [17]. Insignificant amounts of lectins are consumed from unprocessed fruits and vegetables, while raw legumes and whole grains are far more concentrated sources of dietary lectins. Due to their high culinary use around the globe and potential for toxicity, *Phaseolus vulgaris* (common bean) lectins (PHA), and wheat germ agglutinin (WGA) derived from wheat, have arguably received the most attention by researchers [16,18]. Common beans include dark and light red kidney beans, pinto beans, black beans, and white beans. An analysis of raw Canadian legumes measured hemagglutinating activity against rat erythrocytes and found that soybeans showed the highest activity (692.8 HU/mg), followed by common beans (*Phaseolus vulgaris*) (87.69–88.59 HU/mg)*,* lentils (10.91–11.07 HU/mg), peas (5.53–5.68 HU/mg), fava beans (5.52–5.55 HU/mg), and chickpeas (2.73–2.74 HU/mg), respectively [19].

Lectin content may vary with regards to cultivar, cultivation area, and disease susceptibility. Spanish cultivars of chickpeas and fava beans contained greater amounts of lectins, but lesser amounts in soybeans and kidney beans as compared to Canadian pulses [19]. Sun et al. found significant variations in PHA levels among fresh kidney bean cultivars, ranging from less than 200 ug/g to more than 51,200 ug/g. PHA levels appeared to decrease with bean maturity, as concentrations are highest during the growth period for protection [20]. Disease susceptibility and genetic resistance may also play a role in lectin content [21,22].

### 2.3. Effects of Cooking/Processing

Although lectins are fairly resistant to enzymatic digestion in the gastrointestinal tract, they can be removed from foods by various processes (Table 2). For example, soaking, autoclaving, and boiling causes irreversible lectin denaturation. Boiling legumes for one hour at 95 °C reduced hemagglutinating activity by 93.77–99.81% [19]. Adeparusi et al. found that autoclaving lima beans for 20 min eliminated all anti-nutrients except tannins [23]. Boiling of red and white kidney beans, notoriously rich in phytohemoggluttinin (PHA), also resulted in complete elimination of lectins [24]. Microwave ovens on the other hand, are not an effective method for lectin deactivation. Though microwaving destroyed hemagglutinins in most legume seeds, it did not significantly affect lectins in common beans [25]. Additionally, fermentation over 72 h has been demonstrated to destroy almost all lectins in lentils (*Lens culinaris*) [26].

### 2.4. Safety

The safety and overall health effects of dietary lectins has long been a topic of concern among researchers, with some suggesting that they are harmful to health, hence the term ‘anti-nutrients’ [27]. Cases of food poisoning involving raw or inadequately cooked legumes are well documented [28]. For example, in the UK between 1976 and 1989, 50 cases of food poisoning were suspected to be caused by inadequately prepared kidney beans [28]. PHA toxicity, caused by consumption of fresh kidney beans, is also common in China, and affected over 7000 individuals between 2004 and 2013 [20]. In all cases, beans were either consumed raw, soaked, or cooked using temperatures inadequate to destroy PHA. Nonetheless, PHA toxin appears to be eliminated by 10 min of boiling [29]. Mechanistically, lectins and hemagglutinins are resistant to digestion by both host enzymes and bacteria, and therefore pass through the gastrointestinal tract functionally and immunologically intact. Upon arrival into the small intestine, lectins can bind to glycan receptors and glycoconjugates on the luminal surface of the enterocytes [30,31].

In animal models, high doses of isolated legume lectins and raw legume flours have been shown to impair the integrity of the intestinal mucosa by inducing intestinal hyperplasia, altering villus architecture, reducing disaccharidase activity, increasing intestinal permeability and activating the immune system (Table 1) [32,33]. This change in intestinal integrity resulted in compromised nutrient absorption (protein, lipid and vitamin B12) and reduced growth of experimental animals [34,35,36,37]. Nciri and colleagues demonstrated that administration of 300 mg of a raw Beldia bean (white kidney bean) flour for 10 days caused intestinal alterations, distorted jejunum morphology of the microvilli, and reduced enzyme activity in mice [29,38]. Another proposed mechanism of PHA toxicity is intestinal dysbiosis secondary to PHA-induced intestinal damage [37,39]. Clinical human trials using whole (cooked) beans, on the other hand, do not exhibit the same effects as in vitro or in vivo animal models that use isolated lectins and raw bean flours [24].

### 2.5. Human Studies

Clinical human trials of lectin administration are limited. Though lectins from raw legume flours demonstrate adverse effects when administered in isolation in animal models, cooking/autoclaving beans resulted in complete amelioration of PHA and associated erythrocyte agglutination in humans [24]. Contradictory findings may be due to studies which employ animal models, cell cultures, and use isolated lectins. This does not simulate real world scenarios, where lectins are consumed in relatively small amounts with combinations of other foods and bioactive components [40].

In contrast to the anti-nutritional characteristics of lectins initially proposed by many researchers, some evidence suggests that lectins may have therapeutic benefits and could be used as functional foods and nutraceutical agents. Because of lectins’ strong affinity and specificity to glycans, interest lies in their potential as both cancer diagnostic and treatment tools [41]. Current approaches to cancer treatment are often accompanied by harmful side effects due to their poor target specificity, but lectins can identify cancer cells by their secretion of unusual glycan structures. Therefore, lectins are being researched as adjuvants, alongside conventional chemotherapy agents [42,43,44]. Legume lectins isolated from lentils, chickpeas, jack beans, peas and common beans all show anti-proliferative activity against various cancer cell lines, however, human clinical trials are still needed before any conclusions can be made [14].

### 2.6. Conclusions

Overall, research does demonstrate that lectin-rich foods, if not prepared properly, can lead to food poisoning. However, traditional processes such as soaking, sprouting, fermenting, boiling, and autoclaving are all methods that can significantly reduce lectin content. In the case of particularly high-lectin legumes, such as soybeans and kidney beans, boiling or autoclaving is required to eliminate lectins, as reduced cooking temperatures do not significantly affect lectin content. In their whole and cooked form, there is currently no strong evidence from human trials to support the claim that lectin-rich foods consistently cause inflammation, intestinal permeability, or nutrient absorption issues in the general population. Vojdani et al. tested 500 individuals for anti-lectin antibodies and found a range of immunoreactivity between 7.8% and 18% against different lectins, therefore, there may be some individuals who respond to undigested lectins [45]. Of note, legumes and other lectin-rich plant foods are excellent sources of essential amino acids, prebiotic fibers, vitamins, minerals as well as powerful antioxidant and anti-inflammatory compounds [46]. Diets rich in legumes and whole grains are associated with reduced inflammatory biomarkers in both animal and human trials [47,48,49,50]. Until further human clinical trials demonstrate otherwise, the health-promoting effects of lectin-containing foods would seem to far outweigh any possible negative effects of lectins.

## 3. Oxalates

### 3.1. Definition

Oxalate, or oxalic acid, is a substance that can form insoluble salts with minerals, including sodium, potassium, calcium, iron, and magnesium. These compounds are produced in small amounts in both plants, and mammals. All major groups of photosynthetic organisms produce oxalate. It is suggested that plants manufacture oxalate for a variety of functions including calcium regulation, plant protection, and detoxification of heavy metals [51]. In mammals, endogenous oxalate is a metabolite of ascorbate, glyoxylate, hydroxyproline and glycine. Urinary oxalate mostly consists of endogenous oxalate, as opposed to exogenous dietary oxalate. Plant-derived oxalate is available in several different forms; as either water-soluble oxalate (oxalic acid, potassium, sodium and ammonium oxalates) or insoluble oxalate salts (primarily as calcium oxalate) [52]. Soluble (unbound) oxalates can chelate minerals, reducing absorption, or are absorbed through the intestines and colon. Absorbed dietary oxalates are believed to contribute to calcium oxalate kidney stone formation [53]. Insoluble oxalates, on the other hand, are excreted in the feces [54]. Due to their effects on nutrient absorption and possible role in kidney stone formation, oxalates are considered by some to be ‘antinutrients’. Although events of toxicity have occurred in livestock chiefly grazing on oxalate-rich plants [51], a balanced human diet typically contains only small amounts of oxalates [53].

### 3.2. Background

Oxalates are present in many commonly consumed plant foods. Plant foods with the highest oxalate content include spinach, swiss chard, amaranth, taro, sweet potatoes, beets, rhubarb, and sorrel. Raw legumes, whole grains, nuts, baking cocoa and tea also contain oxalate, though in smaller amounts. Distribution of oxalate within a plant can vary. Leaves (spinach, beet greens) are reported to have far greater oxalate content than stalks (rhubarb) or roots (beets, carrots). A distinction should be made between total oxalate, soluble and insoluble oxalate, as excess soluble oxalate has more of an effect on bioavailability and kidney stone formation [54]. Chai and Liebman reported fresh spinach to contain an average of 1145 mg/100 g fresh weight (FW) total oxalate, 803 mg being in the soluble form, and 343 mg being insoluble oxalate [54]. Another group found spinach to contain 978 mg/100 g FW of total oxalate, 543 mg of that being soluble oxalate [55]. Nuts are also reported to be rich in oxalates, ranging from 42 mg/100 g in raw macadamia nuts, to 140, 262, and 469 mg/100 g in roasted peanuts, cashews and almonds, respectively. Soluble content in peanuts and almonds were found to be 108 mg and 153 mg/100 g [56]. Wheat bran contains a somewhat higher amount of soluble oxalate (113 mg/100 g dry weight (DW)), while whole grain products contain much less (13.8 mg in oats, 44 mg/100 g in whole wheat flour) [57].

Raw legumes vary widely in oxalate content. Soybeans contain the greatest amount (370 mg/100 g DW), followed by lentils and peas (168–293 mg/100 g DW), chickpeas (192 mg/100 g DW), and common beans (98–117 mg/100 g DW) [19]. Soluble oxalate in raw chickpeas and lentils is only a fraction of total oxalate [58]. Beet root, another vegetable known for its oxalate content, averages 65 mg/100 g FW of oxalate, with 47 mg being soluble oxalate [54,55]. Differences in total oxalate content is variable among cultivars, season, and growing conditions. For example, among 310 different cultivars of spinach, oxalate concentration ranged from 647.2 to 1286.9 mg/100 g FW, with an average of 984 mg/100 g [59]. Savage et al., on the other hand, found only 266.2 mg/100 g FW in New Zealand grown spinach [53]. Horner and colleagues found over a two-fold difference in oxalate values among 116 cultivars of soy, ranging from 82 to 285 mg/100 g dry weight [60]. Time of harvest can have additional impacts on oxalate concentrations [61]. Research has not demonstrated any differences in oxalate between organic and conventional cultivars [62]. Oxalate values in raw food items are not representative of actual content consumed, as items like legumes and greens are typically cooked prior to consumption. Traditional preparation methods have demonstrated efficacy in significantly reducing oxalate content.

### 3.3. Effects of Cooking/Processing

Like lectins, the cooking, preparation, and processing of food can impact the oxalate content and, therefore, mineral availability of food items (Table 2). Due to oxalate’s solubility in water, wet processing methods such as boiling, and steaming seem to be the most efficient solutions to decreasing oxalate content. Chai and Liebman reported significant reductions of soluble oxalate in vegetables by boiling for 12 min, ranging from 30 to 87% [54]. Spinach and Swiss chard experienced the largest losses (87 and 85%, respectively). Steaming has a lesser impact, though still resulted in losses of 46% and 42% in green Swiss chard and spinach, respectively [54]. Vegetables with lesser exposed surface area and relatively small amounts of oxalate, such as beets, carrots and Brussels sprouts, did not experience similar reductions in soluble oxalate [54]. These results are in agreement with a previous analysis on New Zealand vegetables [53].

Traditional and industrial cooking methods such as soaking overnight and boiling or autoclaving, significantly reduces total and soluble oxalate content in legumes. Cooking lentils on a hot plate for just fifteen minutes reduced soluble oxalate content by 42.6%, and in chickpeas (60 min) by 19.5% [58]. Common beans (cooked for 45 min) experienced a 59.61% loss in oxalate. Even further reductions of 85.7–92.9% were observed in canning (autoclaving) and microwaving of legumes [58]. It has also been found that an overnight soak, followed by a 2-h boil reduced soluble oxalate in red beans by 40.5% [58]. In contrast, there was a 76.9% loss of soluble oxalate in white beans [55]. These differences may be due to variations in genetics, growing conditions, cooking times and exact cooking temperatures. Roasting of peanuts, cashews and almonds did not have any significant impact on oxalate content [56]. In most cases, cooking techniques significantly reduces soluble oxalate, and should therefore enhance mineral availability. Aside from cooking, pairing high-oxalate foods with calcium-rich foods may offset soluble oxalate absorption. A normal calcium diet (800–1,000 mg/day) should be able to offset potential inhibitory effects from dietary oxalates [63].

### 3.4. Safety

Despite evidence of relatively low soluble oxalate concentration in most ‘problematic foods’, dietary oxalate is thought to play a role in hyperoxaluria, a risk factor in the formation of calcium oxalate kidney stones (Table 1). Total dietary oxalate intake is only in the range of 50–200 mg, though in some individuals could be as high as 1000 mg [64]. It has been suggested that dietary oxalate may contribute up to 50% of total urinary oxalate excretion, and that one-third of stone formers hyper-absorb oxalate at a rate of more than 10% total oxalate consumed [65].

### 3.5. Human Studies

A study of 20 healthy men and women found that an oxalate-rich diet (600 mg/day from rhubarb juice) significantly increased urinary excretion from 0.354 to 0.542 mmol/24 h [64]. However, oxalate is not typically consumed every day in such a concentrated form as rhubarb juice, but is, instead, a small fragment in an intricate web of dietary factors. Observing dietary patterns, a prospective analysis from the Nurses’ Health Study (NHS) found only a modest association between dietary oxalate and kidney stone formation after multivariate adjustment [66]. Participants in the highest quintile as compared to the lowest quintile of dietary oxalate, experienced a relative risk of 1.22 for men and 1.21 for older women. Even more significant, in men with lower calcium intake (<755 mg/day), the risk in the highest quintile of dietary oxalate jumped to 1.46. Conversely, in men with calcium intake at or above the median, the multivariate risk dropped to 0.83. Overall, authors concluded that dietary oxalate is not a major risk factor for stone formation [66]. In a more recent NHS I and NHS II analysis, authors again concluded that dietary oxalate had little impact on kidney stone formation, while dietary calcium intake was inversely associated with kidney stone formation [67].

Additionally, dietary potassium, magnesium, and phytate all decrease kidney stone formation through an array of mechanisms [68]. Despite significantly more dietary oxalates (254 mg/day) and oxalate-containing foods such as nuts, vegetables, and whole grains, participants with higher DASH scores have a 40–50% decreased risk of kidney stones [68]. This is perhaps attributed to the protective and synergistic effects of phytate, potassium, calcium, and other phytochemicals all abundant in the DASH dietary pattern. Similar findings regarding the protective role of vegetables on urolithiasis risk were reported by Zhuo et al. [69]. While animal protein consumption was associated with higher kidney stone risk, vegetable and tea consumption were associated with a decreased risk of stone formation. Tea is a rich source of oxalate, yet it is believed that polyphenols and other antioxidant phytochemicals may contribute to the prevention of stone formation [69]. Although there is a connection between calcium oxalate excretion, exogenous (dietary) oxalate, and stone risk, the association may be more complex than once believed.

Gastrointestinal health may also play a role in oxalate absorption and associated health risks. Those with digestive disorders such as inflammatory bowel disease (IBD) have been shown to be at higher risk for calcium-oxalate kidney stones, assumed to be partially caused by oxalate hyperabsorption [70]. Patients with bowel disorders often experience deranged intestinal barrier function, characterized by increased intestinal permeability [70]. Fat malabsorption, secondary to epithelial damage, may also contribute to excess calcium-fatty acid salts, in turn increasing the availability of soluble oxalate [71]. The combination of these factors is theorized to increase oxalate absorption, however, the association between intestinal permeability and oxalate hyperabsorption has yet to be proven. Interestingly, children with autism have demonstrated increased plasma and urinary oxalate levels, but not increased risks of kidney stone formation [72]. This result may be partially explained by increased intestinal permeability and additional dysbiosis found in those with autism spectrum disorders, though is yet to be completely elucidated [73,74]. The gut microbiome, or oxalobiome, may also play a role in reducing dietary oxalates, as bacterial species like *Oxalobacter formigenes* possess oxalate-degrading genes [75]. Nonetheless, human trials using oxalate-degrading probiotics have been mixed, and for the most part, unsuccessful [76,77].

### 3.6. Conclusions

Despite the demonization of oxalate and promotion of a low-oxalate diet in kidney stone patients, more recent observational studies of dietary patterns may prompt a reevaluation of current guidelines. Certain segments of the population do seem to be at greater risk of increased oxalate excretion, and consuming oxalate-rich foods may play a possible role in kidney stone formation, but other factors such as food preparation techniques, calcium intake, endogenous oxalate production, and intestinal health may play a larger role than once thought. Cooking oxalate-rich foods, as well as consuming adequate amounts of calcium and potassium demonstrate efficiency in significantly minimizing available soluble oxalate from dietary sources. Furthermore, oxalate containing foods possess an array of protective, beneficial compounds which may outweigh any possible negative effects of oxalate.

## 4. Goitrogens

### 4.1. Definition

Plant-derived goitrogens are another set of compounds which have received attention among nutrition researchers and health professionals. The term ‘goitrogen’ broadly refers to agents that interfere with thyroid function, thus increase the risk of goiter and other thyroid diseases [78]. Sources of these compounds include medications, environmental toxins, as well as certain foods [79,80]. Glucosinolates, a diverse class of over 120 compounds, are dietary goitrogens found primarily in the *Brassica* family, as well as other plant foods [81]. Upon mastication and ingestion, the enzyme myrosinase (activated in damaged plant tissue and produced by human microflora) converts glucosinolates to a variety of other compounds, including thiocyanates, nitriles, isothiocyanates and sulforaphane [80,81]. Much research surrounding glucosinolates and associated analogues have focused on their potential to prevent cancer, induce phase II detoxification enzymes, induce apoptosis, regulate redox reactions, and inhibit Phase I detoxification enzymes [81,82,83,84,85,86,87]. Despite the potential beneficial effects of glucosinolates, some evidence suggests that goitrin, produced from the glucosinolate precursor, progoitrin, as well as thiocyanate (an indole glucosinolate degradation product), may have adverse effects on the thyroid (Table 1). Early animal and cell models demonstrated goitrin and thiocyanate ions to inhibit the thyroid’s utilization and uptake of iodine [80,88,89].

### 4.2. Background

Vegetables in the *Brassica* genus are the most well-known goitrogen containing foods, although there is an enormous variation of these compounds between species, and even varietals [80]. Kale (*Brassica oleracea acephala* and *B. napus*) and Brussels sprout (B. *oleracea gemmifera*) varietals have been shown to contain the largest amounts of indole glucosinolates and progoitrin, 840 µmol/100 g FW total, and 400.33 µmol/100 g FW total, respectively [80]. However, other studies have found kale to contain very little concentrations of indole glucosinolates and progoitrin [80]. Red Russian kale (*B. napus*) and Siberian kale (*B. napus* ssp *pabularia)* were reported to contain 365.9 µmol/100 g, and 148.1 µmol/100 g FW of progoitrin, respectively. Kale (*B. oleracea acephala*) also contained higher concentrations of glucoraphanin (sulforaphane precursor) than Russian or Siberian species (*B. napus ssp*) [80].

Glucoraphanin is metabolized to sulforaphane and is found to be a potent inducer of Phase II enzymes [82,83,84,85,86,90]. Broccoli, often accused of being high in goitrogens, was actually reported to contain low levels of progoitrin and indole glucosinolates, while being rich in beneficial glucoraphanin [80]. Broccoli sprouts may be an even richer source of glucoraphanin than mature plants, while still containing only negligible amounts of progoitrin [91]. In addition to glucosinolates, resveratrol, isoflavones, and flavonoids may also have goitrogenic effects, though much of the research is based on in vitro or in vivo animal models [92,93,94]. Isoflavones (genistein and daidzein) are found almost exclusively in soy, while resveratrol and other flavonoids are widespread throughout the plant kingdom [95,96]. Millet also contains goitrogenic compounds called C-glycosylflavones, which have been shown in in-vitro models to inhibit thyroid peroxidase (TPO) [97,98].

### 4.3. Effects of Cooking/Processing

Factors such as soil conditions, weather, growing location, use of plant growth regulators or pesticides, pathogen challenges, plant stressors, as well as date of harvest and storage time all can impact glucosinolate content [81,99]. The processing of foods, such as cooking, and fermenting, may lower total glucosinolate concentration (Table 2). However, cooking will also remove beneficial glucosinolates. One study found that steaming broccoli for just 5 min reduced glucoraphanin and total glucosinolate content by 57%, and 51%, respectively [100]. Therefore, it is important to evaluate the current evidence of dietary goitrogens on thyroid and human health, before eliminating or modifying phytonutrient rich plant foods from the diet.

### 4.4. Safety

The evidence published thus far investigating the impacts of dietary goitrogens is mixed and may be more complex than initially thought. “Cabbage goiter” was first observed in rabbits fed a diet consisting almost entirely of cabbage [101]. Later, researchers also observed ‘antinutritional’ effects in rats that were fed high-glucosinolate rapeseed meal and purified rapeseed progoitrin for 30 days [102]. An early human study assessed radioactive iodine uptake following goitrin administration and found that 25 mg (194 μmol) of recrystallized goitrin decreased iodine uptake, though 10 mg (70 μmol) resulted in no inhibition [80]. These results, however, cannot be extrapolated for human health, as they are not representative of a balanced human diet.

Due to the potential inhibitory effects of goitrogens on iodine uptake, populations with underlying iodine deficiency that consume large amounts of goitrogenic foods, may be more at risk than healthy individuals. In rats consuming an iodine-deficient diet containing pure thiocyanate, they experienced significant reductions in thyroxine (T4) levels, as well as reductions in certain proteins and nucleic acids. Adding iodine back to their diet restored levels of thyroxine, reversing the effects of thiocyanate [103]. In contrast, progoitrin-rich rutabaga sprouts had no impact on thyroid function in healthy rats. Adverse effects of iodine deficiency were only pronounced in rats with preexisting hypothyroidism [104].

### 4.5. Human Studies

Human studies investigating the effects of dietary goitrogens in healthy individuals are relatively sparse. Some epidemiological evidence supports an association between goitrogen-containing foods and thyroid dysfunction, though mostly only in the presence of low iodine intake. A study on children found only modest associations between genistein levels and increased thyroglobulin autoantibodies and decreased thyroid volume [105]. In Ethiopian children with iodine deficiency, there was a positive association with consumption of goitrogenic foods (such as taro root, cabbage, Abyssinian cabbage and banana), low levels of iodine in the diet, and lower urinary iodine levels [106]. In a study on pregnant Thai women, higher levels of thyroid stimulating hormone (TSH) were associated with thiocyanate exposure, but only in those with low urinary iodine levels [107]. No associations were found between thiocyanate exposure and thyroid function in mildly iodine-deficient pregnant women [108]. Moreover, consumption of cruciferous vegetables, in combination with low iodine intake, was associated with increased risk of thyroid cancer in women from New Caledonia [109]. A 1.5-fold higher risk of thyroid cancer was observed in a Polish sample who frequently consumed cruciferous vegetables [110]. Other epidemiological studies in the United States have demonstrated an inverse relationship between cruciferous vegetable intake and risk of thyroid cancer [110].

While a small handful of epidemiological studies demonstrate potential concern regarding dietary goitrogens in combination with low iodine, other human studies show no correlations. In a three-year trial of genistein, considered an isoflavone goitrogen, no impacts on thyroid function or health were observed [111]. A review on soy isoflavones arrived at similar conclusions, but still advised soy-consuming individuals taking thyroid medication to increase their dosage of thyroid medication, due to the possibility of decreased drug absorption [94]. Vegans are found to contain slightly higher levels of urinary thiocyanates and lower iodine levels than vegetarians, however no association could be made with thyroid function, based on TSH and T4 levels [112].

Foods exist as complex matrix of compounds, which often have synergistic effects, that have yet to be discovered. In this regard, foods considered to be ‘goitrogenic’ also contain thousands of other bioactive compounds that may be protective against thyroid cancer. According to a case-control study in French Polynesia, a traditional Polynesian diet, rich in cassava and cabbage, was significantly associated with a decreased risk of thyroid cancer when compared to a Western style diet [113]. Zhang et al. found similar negative associations between urinary thiocyanate and thyroid cancer [114]. At the same time, several case-control studies and meta-analysis found no relationship between cruciferous vegetable consumption and thyroid cancer risk [115,116,117].

### 4.6. Conclusions

Overall, most human studies investigating the effects of goitrogenic foods on thyroid health display neutral effects, although some conflicting results are still present. Evidence seems to suggest that suboptimal iodine status may potentiate any negative impacts of dietary goitrogens on thyroid health. Furthermore, progoitrin content amongst the *Brassica* genus varies significantly. Items such as broccoli, Chinese cabbages, bok choi, broccoli sprouts, and some kale varietals generally contain progoitrin and thiocyanate-generating glucosinolates at concentrations far below those likely to cause a physiological effect. In fact, consuming these foods as part of a varied, colorful, plant-based diet should not pose significant risks in healthy individuals, and, conversely, may be of great benefit. In addition to beneficial glucosinolates, cruciferous vegetables provide a plethora of other health-promoting phytochemicals, fiber, and essential vitamins and minerals. For those with thyroid disease, or at higher risk of thyroid disease, long-term daily intake of progoitrin-rich items, like Russian kale, broccoli rabe or collard greens may decrease iodine uptake, and should be cooked with iodized salt to avoid reduced iodine uptake.

## 5. Phytoestrogens

### 5.1. Definition

Phytoestrogens are plant-derived polyphenolic dietary compounds with structural similarities to 17-β-estradiol (E2), the primary sex hormone in females [118]. Due to their similarity to 17-β-estradiol, these bioactive compounds can bind to estrogen receptors (ER), in turn, modulating estrogenic activity. Many tend to have higher affinities for ER-beta than ER-alpha and have a weaker bond than E2 [119]. Phytoestrogens are classified into four phenolic compounds: isoflavones, lignans, stilbenes, and coumestrol [120]. Isoflavones and lignans have received much of the attention, as they are the most relevant with respect to the human diet. Isoflavones are flavonoids found primarily in soybeans, and consist of genistein, daidzein, glycitein, and biochanin A. Lignan phytoestrogens, mostly associated with flaxseeds and other cereals, exist as the glycosides secoisolariciresinol and matairesinol but also include pinoresinol, lariciresinol and syringaresinol [118]. Intestinal microflora are responsible for the conversion to the “mammalian lignans,” enterodiol and enterolactone [121]. Similarly, the microbiome hydrolyzes isoflavone glycosides to their physiologically active aglycone metabolites.

### 5.2. Background

More than 20 isoflavone metabolites have been identified, the most well-studied of which is equol [122]. Equol production varies between populations. It has been found that of Western populations, only about 25–30% are able to convert isoflavones to equol, compared to 50–60% of Asian populations and vegetarians [123]. It is hypothesized that regular consumption of isoflavone-rich foods provides substrates for equol producing bacteria to thrive, if present [123]. There are many suggested health benefits of phytoestrogens, including reduced menopausal symptoms, reduced risk of cardiovascular disease, obesity, metabolic syndrome, type 2 diabetes, cognitive disorders, and various forms of cancer [124,125,126,127,128]. Nonetheless, concerns are frequently raised that soy isoflavones and other phytoestrogens may act as endocrine disruptors and stimulate the growth of estrogen-sensitive cancers [129,130,131,132]. Thus, much debate exists among consumers and clinicians alike, on whether phytoestrogen-rich foods should be included in those with a history or family history of breast cancer.

Phytoestrogens are widespread throughout the plant kingdom, and consumption can vary greatly depending on cultural food preferences. Traditional Asian diets, for example, are estimated to contain 15–50 mg/day of isoflavones, whereas consumption in Western countries is estimated to be only around 2.5 mg/day [133]. This difference can be attributed to the long history of soy products in Asian cuisine. Soy products are one of the richest sources of dietary isoflavones. Whole soybeans contain 103.6 mg/100 g of isoflavones, followed by soy nuts (68.6 mg/100 g), tofu (27.2 mg/100 g), tempeh (18.3 mg/100 g), soymilk (2.9 mg/100 g) and miso soup (1.5 mg/100 g) [134]. Fruits, vegetables, nuts, and other legumes also contain isoflavones, though in significantly lesser amounts [135,136]. Lignans are the second leading source of dietary phytoestrogens, and are ubiquitous throughout plants, though in generally small amounts. Flaxseeds and sesame seeds are reported to contain the greatest amount of lignans, with 379.4 mg and 8.00 mg/100 g respectively [134]. Nuts were found to contain between 0.025 mg and 0.198 mg/100 g [137]. Lignans, in general, were found to be negligible in legumes, fruits, vegetables and cereals (< 0.01 mg/100 g). Exceptions were noted for garlic, olive oil, winter squash, dried apricots, dried dates, dried prunes and multigrain bread [134].

### 5.3. Effects of Cooking/Processing

As previously stated, dietary phytoestrogen glycosides must first be transformed to aglycones by glucosidases before they can be utilized by humans [122,123,138]. Glycosides can be hydrolyzed via intestinal glucosides, intestinal bacterial glucosides, as well as through various processing methods [123,139,140,141]. Boiling and steaming led to significant increases in beta-glucosides and aglycones, though pressure steaming resulted in the greatest amounts (Table 1) [139]. Fermentation by *Lactobacillus* and *Bifidobacteria* also results in increased aglycone content [141]. Bau et al. found that by fermenting soymilk for 30 h with kefir culture, glycitin and daidzin were completely hydrolyzed into aglycones, while 89% of genistin was bioconverted [140]. Consuming traditionally fermented soy products, such as Korean cheonggukjang, Japanese natto, and Thai Thua, may further enhance isoflavone bioavailability, though more human trials are necessary [122].

### 5.4. Safety

Phytoestrogens have received a large amount of attention over the past few decades, particularly because of their potential estrogenic effects (Table 1). For this reason, much research has examined possible benefits of phytoestrogens on menopause symptoms, although results have been mixed [137]. A recent systematic review and meta-analysis concluded that phytoestrogen supplementation resulted in significantly greater reductions in hot flashes as compared to placebo, but did not significantly impact the Kupperman Index, an index which included 11 symptoms of menopause [142]. Another meta-analysis found similar benefits in the ability of soy isoflavones to improve hot flashes, as well as vaginal dryness score [143]. Chen and colleagues concluded in a recent literature review that isoflavones reduced hot flashes, attenuated bone mineral density (BMD) loss in the lumbar spine and may have potential benefits on blood pressure and glycemic control [144].

Nonetheless, a recent Cochrane review was unable to conclusively state that phytoestrogens are effective for reducing menopausal symptoms due to the heterogeneity of studies, and individual variability in metabolism and absorption of isoflavones [145]. An exception was noted for genistein supplementation of 30–60 mg/day, which reliably demonstrated a benefit for hot flash frequency [145]. The heterogeneity in results may be partially explained by equol. An observational study of 365 peri- and post-menopausal women, found that equol producers in the highest quartile of daidzein intake were 76% less likely to report vasomotor symptoms than those in the lowest intake quartile. No associations were found between daidzein intake and vasomotor symptoms in equol nonproducers [146]. Equol supplementation may also be of benefit to non-producers. A 12-week double-blind RCT found that equol supplementation (10 mg/day) improved mood-related symptoms, even in non-producers. Those that received 10 mg three times daily demonstrated significantly better outcomes in all measures [147]. A meta-analysis of equol supplementation also revealed significant improvement in hot flash severity, both in equol producers and non-producers [148].

Another primary concern regarding phytoestrogens is due to their possible endocrine-disrupting effects [129]. Due to the rising rates of soy-based infant formulas, developing babies and infants may be most at risk. Serum genistein concentrations are 10–50-fold higher in soy-formula fed infants than in Asian adults, and 100–700-fold higher than US adults [149]. Nonetheless, the biological significance of increased phytoestrogen exposure in infants is yet to be determined [150,151]. Collective findings in adults have not identified conclusive evidence that soy food or isoflavones adversely affect thyroid function in euthyroid or iodine-replete individuals [94].

The other common concern surrounding soy and phytoestrogen intake is increased risk of estrogen-sensitive breast and uterine cancer [132]. Thus far, no evidence has demonstrated a link between phytoestrogen-rich diets and estrogen-sensitive malignant growths. In contrast, soy consumption may actually be associated with reduced risk of breast cancer incidence, recurrence and mortality [132,152].

### 5.5. Human Studies

Studies investigating the specific potential impact on female reproductive health are mixed. A systematic review and meta-analysis concluded that isoflavones have no effect on endometrial thickness or breast density [153]. Another meta-analysis of pre- and postmenopausal women found isoflavones to have only a weak effect on the hypothalamic-pituitary-gonadal axis [154]. In premenopausal women, soy isoflavone consumption had no effect on circulating estradiol, estrone or sex hormone binding globulin (SHBG). Follicle stimulating hormone (FSH) and luteinizing hormone (LH) concentrations were significantly reduced, and menstrual length increased by 1.05 days. However, once bias was accounted for, changes were no longer significant [154]. In postmenopausal women, no statistically significant effects were noted for circulating total estradiol, estrone, SHBG, FSH, LH or TSH, though soy increased total circulating estradiol non-significantly [154]. Women that were fed soy-formula as an infant reported slightly longer menstrual bleeding times (0.37 days), and greater discomfort during menstruation than cow milk fed infants [155]. Another study conducted on Korean girls with central precocious puberty (CPP) found a positive association between elevated serum isoflavones and risk of CPP [156]. As soy-based formulas are also known to contain pesticide and glyphosate residues, effects of soy cannot be attributed to phytoestrogens alone [157].

Despite concerns over estrogen’s endocrine disrupting effects, estrogen is (E2) is proposed to play a role in protection against cardiovascular disease (CVD), and the ensuing increased risk of CVD post-menopause once E2 levels decline [158,159]. Due to the structural similarities to E2, phytoestrogens have also been investigated for possible cardiovascular benefits. Epidemiological evidence suggests potential protective effects of phytoestrogens, particularly in Asian populations with high isoflavone intake from soy products [160]. A positive relationship has been found between isoflavone intake, endothelial function and reduced lower carotid atherosclerotic burden [161]. Ferreira and colleagues also found that higher isoflavone intake was independently associated with lower risk for subclinical CVD in menopausal women [162]. Results from experimental studies using phytoestrogens for CVD prevention and treatment have been mixed, but generally positive. In one study, soy isoflavones in combination with probiotic resistant starch or probiotics (*L. acidophilus, B. bifidus and LGG),* was shown to significantly decrease total and LDL cholesterol, independent of isoflavone bioavailability [163]. Another study using 15 g of soy protein with 66 mg isoflavone daily for 6 months resulted in significant reductions in systolic blood pressure (SBP). The reductions in SBP led to a 27% reduction in 10-year coronary heart disease risk, a 37% reduction in myocardial infarction risk, a 24% reduction in cardiovascular disease and 42% reduction in CVD death risk [164]. A meta-analysis of 17 RCTs suggested that isoflavone-containing soy products can modestly, but significantly improve endothelial function, as measured by flow mediated dilation (FMD) [165]. Finally, several studies have suggested that genistein significantly improves FMD, reduces endothelin-1 levels, and induces nitric oxide-dependent vasodilation to a similar extent of estrogen [166,167,168].

Soy-based and phytoestrogen-rich products have also been proposed for the prevention of certain cancers, including breast, prostate, endometrial, and colorectal cancer [119,169,170,171,172]. Some studies, however, have suggested that soy isoflavone intake is associated with significantly reduced breast cancer risk only in Asian populations, but not in Western populations [173,174,175]. Ingestion of phytoestrogens and soy may also offer significant protection against prostate cancer. A recent meta-analysis from the University of Illinois found that total soy food, genistein, daidzein, and unfermented soy food to be significantly associated with reduced advanced prostate cancer risk [176]. In another meta-analysis, soy isoflavone supplementation led to a significant reduction in prostate cancer diagnosis in those with an identified risk [177]. No reductions in PSA levels or steroid endpoints were observed.

The benefits of phytoestrogens may be due to their anti-inflammatory and antioxidant properties [178]. Data from the 1999–2010 NHANES revealed an inverse associated with urinary phytoestrogens and serum C-reactive protein (CRP), a marker of inflammation [176]. These results should be interpreted with caution however, as an increased intake of phytoestrogens in Western cultures may be evidence of an overall healthy diet, rich in a variety of other nutrients and bioactive compounds that reduce CRP levels [179].

### 5.6. Conclusions

Overall, the evidence surrounding phytoestrogens within the currently published literature is still mixed, with a large amount of heterogeneity between studies. The microbial makeup of the gut, bio-individuality, and the phytoestrogen source all play a significant role in the decision to include phytoestrogen-rich foods in one’s diet. Supplementation using isolated isoflavones may be beneficial for some populations but may pose increased risk for others. Babies and infants are at higher risk of the endocrine-disrupting potential because of their small size and underdeveloped digestive tract. With that said, epidemiological and observational data suggests that including phytoestrogen-rich foods as part of a varied, plant-based diet should not be of concern, but may be beneficial. Additionally, phytoestrogen-containing foods such as legumes, grains, seeds, nuts, fruits, and vegetables, are rich sources of essential vitamins, minerals, fiber and other health-promoting phytochemicals.

## 6. Phytates

### 6.1. Definition

Phytate, also known as phytic acid or *myo*-inositol hexaphosphate (IP6), is another commonly considered “anti-nutrient” widely distributed in amongst the plant kingdom. It primarily serves as storage for plant phosphate, as an energy source, and antioxidant for germinating seeds [180]. Phytate is produced during seed development and can account for 60–90% of total phosphorus content in cereal grains, nuts, seeds, and legumes [181]. Structurally, phytate (IP6) is made up of six phosphate groups, attached to an inositol ring, with the ability to bind up to 12 protons total. These phosphate groups act as strong chelators, readily binding to mineral cations, particularly Cu2+, Ca2+, Zn2+, and Fe3+ [182]. These complexes are insoluble at neutral pH values (6–7), and cannot be digested by human enzymes, thus could decrease mineral bioavailability in high-phytate, homogenous diets [12]. Low-income, developing countries that rely predominantly on grains and legumes as dietary staples are of special concern for zinc deficiency and/or insufficiency [183]. The chelating properties of phytate also allow it to act as an antioxidant, lending possible protective traits [180]. Ensuring an appropriate phytate to mineral ratio minimizes the negative effects of phytate on mineral absorption in vulnerable populations.

### 6.2. Background

Phytate is found in a wide array of plant foods, with the highest concentrations occurring in cereals, legumes, nuts, seeds and pseudocereals [182]. In cereals, phytate is mainly found in the outermost layer, and in legumes is found within the endosperm and cotyledons [180]. Reported daily intake of phytate for vegetarians and other rural inhabitants in developing countries is estimated to be 2000–2600 mg, while mixed diets may contain as low as 650 mg of phytate [184]. Growing methods, seasons, and cultivars can have a significant impact on phytate content [185,186,187]. Shi et al. reported phytate content in Canadian grown peas, lentils, fava beans, chickpeas, and common beans to be 9.93–12.40 mg/g, 8.56–17.1 mg/g, 19.65–22.85 mg/g, 11.33–14 mg/g and 15.64–18.82 mg/g, respectively. Soybeans contained the highest amount, at 22.91 mg/g [19]. However, Wang and colleagues reported Canadian lentil, chickpea and bean cultivar values to be much less, containing 7.2–11 mg/g, 9.6–10.6 mg/g and 9.9–13.8 mg/g, respectively [188,189]. Split varieties of lentils and peas contain more phytate, since much of the hull is lost during processing [19]. Unprocessed cereals generally have similar phytate value to that of legumes, though processed cereals contain significantly less. For instance, wild rice contains between 12.7 and 21.6 mg/g, but polished rice, only 1.2–3.7 mg/g [184]. Upon processing, phytate content can be significantly reduced in many grains, seeds, and legumes.

### 6.3. Effects of Cooking/Processing

Processing techniques such as soaking, fermentation, sprouting, germinating, and cooking can significantly alter phytate content in grains and legumes, allowing for increased mineral availability (Table 2). Cooking of legumes for 1 h at 95 °C resulted in up to a 23% loss in yellow split peas, 20–80% loss in lentils, and 11% loss in chickpeas. Only a marginal reduction of 0.29% was noted in black beans [19]. Utilizing the natural phytases present in cereals and legumes has proven to be a valuable tool in reducing phytate. Phytases are enzymes capable of hydrolyzing phytate. Soaking seeds in fresh water, a traditional preparation method, reduced phytate values in millet, maize, rice, and soybean by 28, 21, 17, and 23%, respectively [190]. No IP6 was found in the soaking water, implying that the phytate was hydrolyzed by endogenous cereal phytases. Although soaking reduced phytate, it also resulted in significant losses of iron and zinc in the soaking water. For this reason, soaking did not lower the phytate/iron ratio, and only had minor impacts on the phytate/zinc ratio [190]. Mineral loss could be partially mitigated by cooking rice in the soaking water, as the seeds will ‘recover’ the leached minerals. Germination of foods can further reduce phytate, as endogenous phytases are activated to free the phosphate from phytate to be used for energy. Germinating chickpeas and pigeon peas reduced phytate concentrations by over 60%, while still preserving mineral content [191,192].

Fermentation, such as the natural leavening of bread, has also been found to significantly reduce phytate. It is elucidated that along with activity of bacterial phytases, lactic acid bacteria activate endogenous cereal phytates by lowering the pH of the dough to ~4–5. Slight acidification with lactic acid produces similar results [193]. Additional research by Castro-Alba et al. demonstrated that inoculation of quinoa, amaranth, and canihua with *L. plantarum* 299v reduced phytate concentrations by 47–51%, 12–14%, and 25–27%, respectively. Accessibility of iron, zinc and calcium was also increased in the fermented flours [194]. Furthermore, *L. plantarum* species from supplements (*L. plantarum* 299v), or from fermented vegetables (*L. plantarum* spp.), have been found to improve iron bioavailability from high phytate meals [195,196]. In a study performed by Scheers et al., iron absorption increased from 13.6 to 23.6% in the low phytate meal, and 5.2 to 10.4% in the high phytate meal, when eaten alongside fermented vegetables. Zinc absorption changed minimally [195]. The exact mechanism is unknown but may be due to an increase in ferric iron [195,197]. Supporting microbiome health through the consumption of fermentable fibers and other prebiotics also lowers cecal pH values, allowing for an increased solubility of zinc and iron [198,199,200]. High-phytate foods such as beans are rich in fermentable fibers, which have a beneficial effect on cecal pH by increasing SCFA production [201,202]. This effect may lend insight into the phenomenon of phytate adaption, in which non-heme iron absorption can be partially negated by the consumption of a high-phytate diet [203].

### 6.4. Safety

As discussed, phytate is viewed as an ‘anti-nutrient’ because it can chelate iron, calcium, and zinc, limiting absorption of these minerals (Table 1). Chelation, however, is dependent on the proportion of phytate to metal ions, as well as pH [204]. The ideal molar ratio of phytate to iron is ~0.4, with an inhibitory effect in ratios above 1. For zinc, ratios higher than 15 may inhibit absorption rates, with optimal ratio of below 5. Calcium absorption has been shown to be impeded by molar ratios above 0.17 [182].

### 6.5. Human Studies

Many studies support the hypothesis that phytate negatively impacts zinc bioavailability [205,206], however a study on young children, aged 8–50 months, found phytates to not have a discernable effect on zinc absorption [207]. An increase of 500 mg/day of dietary phytate led to less than a 0.04 mg/day reduction in zinc absorption. The largest variance in absorption rates occurred based on age, height and weight [207]. The relationship between dietary phytate and iron bioavailability may be more complex than that of zinc. Even after removal of 90% of IP6 in sorghum flour through phytase treatment, no improvement in iron bioavailability was observed [208]. Removing fiber was found to have a more significant impact on iron absorption, demonstrating an independent effect of fiber in phytate-rich foods. Also, despite higher phytate concentration, animal models have found whole-wheat flour to result in greater iron absorption than refined white flour [209]. Nonetheless, due to phytate’s overall impact on zinc and iron absorption, it is recommended that DRIs be increased for these minerals to account for bioavailability values [210,211]. The addition of complementary foods high in ascorbic acid (vitamin C) may allow consumers enjoy the benefits of phytate-rich foods, while offsetting phytate’s inhibitory effect on mineral absorption [212].

Although phytates are viewed by many in a negative light, they may actually act as beneficial antioxidants by their ability to chelate excess iron, thereby preventing damaging Fenton reactions from taking place [204]. Fenton reactions are oxidative reactions involving iron and hydrogen peroxide, producing hydroxyl radicals and other reactive oxygen species (ROS) [213]. Not only can excess iron contribute to ROS through the Fenton reaction, but research has linked heme-iron to microbial dysbiosis, hyperproliferation of colon cells and altered intestinal barrier function [214]. Since only a small amount of heme is absorbed in the small intestine, up to 90% may reach the colon [214]. When eaten alongside heme-rich foods, phytate acts as ‘nature’s iron regulator’, attenuating possible heme-induced damage. Animal studies have demonstrated a protective role of IP6 against iron-induced lipid peroxidation in the colon [215]. However, human trials validating phytate’s elucidated antioxidant effects are limited. In one randomized cross over trial, Sanchis et al. reported significant reductions (~25%) of advanced glycation end-products (AGEs) in patients with type 2 diabetes mellitus supplemented with 1 g of IP6 [216]. Considering the deleterious effects AGEs have on microvascular and macrovascular function in type 2 diabetes (T2DM), dietary phytates could be promising tools in the treatment of T2DM.

Phytate may also possess other beneficial effects, yet much of this research is still in its infancy. The mechanisms of action of IP6 include enhanced immunity, inhibition of inflammatory, cytokines, caspase modification, regulation of phase I and II enzymes, and decreased cell proliferation [180,215]. IP6 has also been shown to decrease kidney stone risk [217], dental calculi [218], osteoporosis risk [219], and help prevent age-related cardiovascular calcification [220,221]. Furthermore, adequate consumption of phytate-rich foods was found to prevent abdominal aortic calcification in patients with chronic kidney disease [180,222]. Future research is needed to identify the exact physiological mechanisms behind phytate, but research thus far supports the inclusion of phytate-rich foods into a balanced diet.

### 6.6. Conclusions

Since its discovery, the role of phytate in human nutrition has been a controversial topic. On one hand, phytate may decrease the bioavailability of essential minerals, while on the other hand, acts as a potent antioxidant. Phytates should not significantly impair mineral status when included as part of a diverse and balanced diet, especially if using traditional processing methods such as soaking, germinating, fermenting, and cooking. Consuming complementary foods rich in ascorbic acid and certain probiotic bacteria could also have beneficial impacts on mineral absorption from high-phytate meals. Overall, by consuming a colorful, plant-based diet, the benefits of phytate containing foods to human health exceed the impacts on mineral absorption.

## 7. Tannins

### 7.1. Definition

Tannins are a broad class of polyphenol compounds of high molecular weight (500–3000 Daltons) ubiquitously present in commonly consumed plant foods and are responsible for the astringent taste of many fruits and beverages [223]. They can be chemically classified into two groups: hydrolysable tannins and condensed tannins (also known as catechin tannins, flavanols, or proanthocyanidins). Hydrolysable tannins, including gallotannins and ellagitannins, are selectively found in the diet. Condensed tannins, or proanthocyanidins, on the other hand, are the most abundant plant-derived polyphenols in the diet and include catechin, epicatechin (EC), epigallocatechin (EGC), epicatechin-3-gallate, and (-)-epigallocatechin-3-gallate (EGCG) [224].

Due to their phenolic nature, tannins are chemically reactive, forming intra- and inter-molecular hydrogen bonds with macromolecules like proteins and carbohydrates. This lends to their role in plant defense, as well as to their antioxidant, anticarcinogenic, immunomodulatory, detoxifying, and cardioprotective activities [225,226,227,228,229]. Tannins may act as antioxidants by scavenging free radicals, although their ability to act as chelators have also been reported to inhibit the absorption of dietary minerals such as iron, copper, and zinc [230]. The elucidated ‘anti-nutritional’ effects of dietary tannins have been suggested as a contributor to iron-deficiency anemia, particularly in developing and low-income countries who rely on tannin-rich foods [231]. Other studies suggest that iron status and absorption is not significantly affected by dietary tannin intake and is found to be highly variable between individuals [227,232].

### 7.2. Background

Tannins, specifically proanthocyanidins or catechins, are one of the most abundant secondary plant metabolites, found in cocoa beans, tea, wines, fruits, juices, nuts, seeds, legumes and cereal grains [225]. Dark and baking chocolate contains the highest amounts of proanthocyanidins (828–1332 mg/100 g) [225]. A Danish study found fruits with the richest concentrations of catechins included black grapes (203.9 mg/kg FW), apples (71.1–115.4 mg/kg FW), apricots (110 mg/kg), plums (61.9 mg/kg), cherries (117.1 mg/kg), all edible berries (11.1–187.4 mg/kg), pears (30.6–85 mg/kg), cranberries (42 mg/kg), and peaches (23.3 mg/kg) [233]. Nuts (almonds, walnuts, pecans, and pistachios), common beans and some cereals, such as sorghum, also contain notable amounts of catechins [233]. Darker beans, such as dark red kidney beans, have been shown to contain more catechins than lighter beans [233].

Tea and wine are rich sources of catechins. Arts et al. found that of the red wines tested, catechin values were between 27.3 and 95.5 mg/L [234], though others have cited values as high as 300 mg/L [225]. Content in tea has been found to be between 100 and 800 mg/L in green tea, and 60–500 mg/L in black tea [225]. Tea is the predominant source of epigallocatechin gallate (EGCG), a powerful and well-studied antioxidant [235,236]. Ceylon has been reported to contain the most EGCG (128–229 mg/L) [234]. Ellagitannins, a class of hydrolysable tannins, are found in a limited number of fruits and nuts, including walnuts, pecans, berries and pomegranates [225].

### 7.3. Effects of Cooking/Processing

Cooking and processing may decrease total catechin content in some foods (Table 2). Arts et al. reported reductions in rhubarb, broad beans and pears by 28, 58 and 26%, respectively [233], although a majority of catechin-rich foods, like fruits, are consumed raw. Removing the skins from nuts may reduce phenolic content by up to 90% [230,233]. Catechin content in tea increases with the amount of tea used and with increased infusion time, however catechin concentrations and antiradical activity seem to peak at 4–5 min of brew time [234,237]. Tannin content in foods and tea can be influenced by region, variety, processing methods, and storage time [233,234,238]. Polyphenols were found to vary significantly between agricultural methods, though not as much as between cultivars [239,240,241].

### 7.4. Safety

Despite their ubiquitous nature in many nutritionally dense plant foods, some researchers and clinicians have deemed tannins as antinutritional factors due to their potential to reduce iron absorption (Table 1) [230,231,242]. Early animal studies reported tannins to cause depressed growth and egg production in poultry, when fed at levels of 0.5–2% of feed [242]. In weaning pigs, consumption of 125, 250, 500, or 1000 mg tannic acid/kg in feed resulted in a significant drop in hemoglobin, and depletion of serum iron concentrations. However, erythrocyte counts, hemoglobin and hematocrit decreased similarly in the control group to that of the 125, 250, and 500 mg/kg diet groups [243]. Other animal studies using condensed tannins (more commonly found in the human diet) have not found any significant impacts on iron status [244].

### 7.5. Human Studies

The aforementioned concentrations are far greater than regularly consumed through a diverse diet. Delimont and colleagues found that 4-weeks of condensed tannin supplementation (1.5, 0.35 and 0.03 g 3 times/day) had no impact on iron bioavailability or status in premenopausal women [245]. Tea, one of the richest sources of dietary tannins, may inhibit iron absorption when consumed directly with a nonheme iron-rich meal. In a study of healthy adults, iron absorption was decreased by 37% when tea was consumed with an iron-fortified porridge, however, was not affected when tea was consumed an hour after the meal [246]. Other factors, such as gender and baseline iron status, may also influence the impact of tannins on iron parameters. In a study investigating the effects of green and black tea on iron status of omnivores and vegetarians, 1 L of black tea/day for four-weeks (with meals) resulted in significantly lower ferritin levels only in omnivorous females, but no effects were observed in omnivorous males [247]. Green tea had no influence on ferritin levels in omnivorous and vegetarian females. In females with low baseline ferritin (<25 μg/L), both green and black tea significantly reduced ferritin levels [247].

Tannins are not consumed alone, but in combination with thousands of other bioactives, including ascorbic acid. Potential inhibitory effects of tannins may be offset by the inclusion of 30 mg of ascorbic acid [248,249,250]. This may explain why human epidemiological studies investigating iron deficiency anemia are unable to demonstrate any correlations between dietary tannin intake and iron-deficiency anemia. Of 2593 French subjects, serum-ferritin concentrations were not related to tea consumption, independent of strength, infusion time or time of tea drinking [251]. A cross-sectional analysis of 1605 healthy adults also found that tea consumption did not significantly increase risk for iron deficiency or iron-deficiency anemia [252]. Similar findings were also shown by Root et al. in adults from rural China [232]. A systematic review by Speer et al. concluded that total polyphenol intake did not interfere with iron status but did improve inflammatory biomarkers in participants [253]. The review included a limited number of studies, but it speaks to the numerous demonstrated health benefits of tannins and tannin-rich plant foods.

Although the ‘anti-nutritional’ effects of tannins are debatable and highly variable, evidence supporting the many health benefits of tannins are widespread [225,228,254]. Dietary intake of polyphenols is associated with a decreased risk of T2DM, metabolic syndrome, risk of ischemic stroke, non-fatal cardiovascular events risk, and risk of atherosclerotic vascular disease [254]. The Takayama study, consisting of over 29,000 Japanese individuals, found significantly lower CVD mortality in subjects with the highest polyphenol intake, as compared to those in the lowest quartile [255]. Inverse associations also existed for mortality from digestive diseases. Polyphenols in this population were mainly derived from beverages such as green tea and coffee [255,256]. Consumption of proanthocyanidin-rich foods has been shown to reduce the risk of chronic kidney insufficiency and renal disease [257]. Proanthocyanidins are believed to exert their renal and cardioprotective effects by reducing oxidative stress and improving endothelial function [258,259,260]. A randomized crossover study found that drinking 3 cups black tea resulted in immediate improvement in brachial artery FMD in healthy subjects [261].

Tea catechins and ellagitannins may lower CVD risk by upregulating Nrf2 [nuclear factor erythroid 2 (NF-E2) p45-related factor 2] [262,263]. Nrf2 is a key transcription factor responsible for the body’s detoxification and antioxidant defense systems [229]. Ellagic acids, present in raspberries, strawberries, pomegranates, and nuts have demonstrated anticarcinogenic effects in vivo. Animal models suggest that ellagic acid may modulate phase I and phase II enzymes by lowering or inhibiting cytochrome P450 enzymes, and inducing glutathione-s-transferase, UDP and NAD(P)H-quinone reductase activity [264,265,266]; however, human clinical data indicating similar effects has not been demonstrated.

Furthermore, flavanol-rich foods, such as fruits, vegetables, and cocoa demonstrate positive effects on cognition, executive function, and even mood, although exact mechanisms are yet to be elucidated [267,268,269,270]. Neshatdoust et al. observed significant improvements in cognitive performance and increases in brain-derived neurotropic factor (BDNF) levels after an 18-week intervention of high-flavonoid fruits and vegetables (>15 mg/100g) [267]. Another intervention utilizing a high-flavanol cocoa beverage (494 mg total flavanols) resulted in significantly higher brain-derived neurotropic factor (BDNF) levels in older individuals, when compared to the low-flavanol cocoa drink (23 mg total flavanols) group [267]. The CoCoA study, an 8-week supplementation with a high-flavanol cocoa drink (993 mg flavanol), reduced measures of age-related cognitive dysfunction. Significant improvements in insulin resistance, blood pressure and lipid peroxidation were also observed in the high flavanol (993 mg) and intermediate flavanol group (520 mg), suggesting insulin modulation as a possible mechanism [268]. Grassi et al. found that consumption of high-flavanol dark chocolate ameliorated vascular impairment after sleep deprivation and improved working memory performance [271], indicating that cognitive improvements may be due to effects of flavanols on blood pressure and peripheral and central blood flow.

Flavanols may additionally act as prebiotics, positively influencing the gut microbiota, in turn alleviating neuroinflammation and balancing serotonin metabolism [256]. Ingestion of a high-cocoa flavanol drink (494 mg cocoa flavanols) significantly increased *Bifidobacteria* and *Lactobacilli* populations, while at the same time significantly decreasing *Clostridia* counts, when compared to the low-cocoa flavanol (23 mg) drink [272]. Significant reductions in plasma triacylglycerol and C-reactive protein concentrations were also linked to the changes in microbial counts [272]. Being that many polyphenols are metabolized by gut microbiota [256], individual microbial composition and dietary habits may influence both the bioavailability and physiological effects of flavanol-containing foods.

### 7.6. Conclusions

Tannins are highly bioactive compounds which are widely found in plant foods and beverages, including berries, apples, stone fruit, cocoa, legumes, whole grains, tea as well as many others. Although some studies have found that tannins may interfere with iron absorption when consumed in isolation, other studies investigating whole diets demonstrate otherwise. Harmful (and even beneficial) effects of an individual, isolated compound or phytochemical are often quite different than when the same compound is within the complex food matrix. For this reason, epidemiological evidence has not demonstrated any correlation between iron status and flavanol intake. Ascorbic acid, present in many tannin-rich foods, may further enhance the absorption of non-heme iron. Nonetheless, some studies still advise that those with low iron stores, especially females, consume tannin-rich beverages, such as tea, after or in-between meals to avoid potential effects on iron absorption. Overall, evidence suggests that the many health benefits of consuming a diverse, plant-based diet, rich in polyphenol and bioactive containing foods and beverages, far outweighs the potential impact of tannins on iron status.

## 8. Limitations

There are limitations to this narrative review that should be noted. First, human clinical trial research investigating the effects of antinutritional compounds in whole food form are limited, and in some cases do not always arrive at clear-cut conclusions. In place of clinical trials, epidemiological and observational studies must be used, though are typically limited in their applicability due to uncontrolled variables. Second, much of the research on antinutritional components are performed using isolated compounds in animal models, which are not representative of a balanced diet. Research limitations are further compounded by the often synergistic nature of food, effects of cooking and processing, as well as the bio-individuality of study participants. More research which takes these variables into consideration are needed before definitive conclusions can be made regarding the ill-effects of these compounds in their whole food form.

## 9. Overall Conclusions

The purpose of this review was to assess whether there is considerable clinical data to warrant certain compounds in plants to be positioned as ‘anti-nutrients’ in the sense that they block the absorption or assimilation of essential nutrients or, in some way, interfere with physiological function of an organ. The summary of our findings would suggest the following:

(1)Of the compounds reviewed, there are indications that when given in the diet in what would be considered moderate to high quantities, or when administered in isolation, they may exert effects that would be detrimental or impair the body’s reserves or function in some way. There may be some individuals who are more susceptible to these effects for various reasons.(2)These compounds are rarely ingested in their isolated format as we know from how these foods are traditionally consumed. Plant-based diets which contain these compounds also contain thousands of other compounds in the food matrix, many of which counteract the potential effects of the ‘anti-nutrients’. Therefore, it remains questionable as to whether these compounds are as potentially harmful as they might seem to be in isolation, as they may act differently when taken in within whole foods that are properly prepared. Cooking and application of heat seems to be essential for the activation of some of these compounds.(3)In some cases, what has been referred to as ‘anti-nutrients,’ may, in fact, be therapeutic agents for various conditions. More exploration and research are required to know for certain.

## Figures and Tables

**Table 1 nutrients-12-02929-t001:** Plant Compounds, Food Sources, and Their Suggested Clinical Implications.

‘Anti-nutrient’	Food Sources	Suggested Clinical Implications
Lectins	Legumes, cereal grains, seeds, nuts, fruits, vegetables	Altered gut function; inflammation
Oxalates	Spinach, Swiss chard, sorrel, beet greens, beet root, rhubarb, nuts, legumes, cereal grains, sweet potatoes, potatoes	May inhibit calcium absorption; May increase calcium kidney stone formation
Phytate (IP6)	Legumes, cereal grains, pseudocereals (amaranth, quinoa, millet), nuts, seeds	May inhibit absorption of iron, zinc and calcium; Acts as an antioxidant; Antineoplastic effects
Goitrogens	*Brassica* vegetables (kale, Brussels sprouts, cabbage, turnip greens, Chinese cabbage, broccoli), millet, cassava	Hypothyroidism and/or goiter; Inhibit iodine uptake
Phytoestrogens	Soy and soy products, flaxseeds, nuts (negligible amounts), fruits and vegetables (negligible amounts)	Endocrine disruption; Increased risk of estrogen-sensitive cancers
Tannins	Tea, cocoa, grapes, berries, apples, stone fruits, nuts, beans, whole grains	Inhibit iron absorption; Negatively impact iron stores

**Table 2 nutrients-12-02929-t002:** Preparation tips for reducing ‘anti-nutrients’.

‘Anti-nutrient’	Food Preparation that Reduces	Food Preparation that Increase
Lectins	Soaking, boiling, autoclaving, germination, fermentation	Roasting, baking
Oxalate	Soaking, boiling, steaming, pairing with high calcium foods	Roasting, grilling, baking, low-calcium diet
Phytates	Soaking, boiling, germination, fermentation	*n*/a
Tannins	Cooking, peeling skins of fruits and nuts	*n*/a
Phytoestrogens	*n/a*	Boiling, steaming, fermenting (increases aglycone content)
Goitrogens	Steaming, boiling

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
