# Peer review of "Is There Such a Thing as “Anti-Nutrients”? A Narrative Review of Perceived Problematic Plant Compounds"

_nutrients, 2020, doi:10.3390/nu12102929_

Round 1
Reviewer 1 Report
This review by Petroski and Minich reports a detailed examination of the so called “antinutrients” elements that occur in dietary vegetables in an attempt to evaluate critically the pro and cons of a large intake of vegetable, fruits and grains as required in diets recommended for reducing the risk of chronic diseases.
Overall the review is of interest and the various antinutrients are described in a clear way so as the food processing/cooking strategies to reduce their impact while leaving the potential beneficial effects of the plant sources
Of course since many papers and studies are briefly described it is not always easy to follow the evidence presented. I would encourage the authors to find for each antinutrient a pictorial way to represent those they judge as the most significant and reliable studies. Also, it seems that most of human studies are controversial or at least did not reach a clearcut result. The Aus may comment more on this aspect. Moreover, the conclusion paragraph which is most useful to summarize the content of the review and appreciate the still controversial issues may be expanded. I found very useful summary offered by tables 1 and 2
Minor points
The format of the references is unusual and different from what required by the journal style (my visualization shows roman ordinal numbers)
Line 167 oxalates form salts with cations not properly bind them
Line 191 … 978 mg apparently /100 g is missing
Line 356 and other use “pure” rather than “isolated”
Author Response
- We attempted to make the manuscript easier to follow by reducing some sections and streamlining the studies presented, with a primary focus on the clinical trials.
- A limitations paragraph was added towards the end in order to present issues where there could be controversy.
- References have been re-done with numerical sequences.
- The point about oxalates was corrected.
- The measurement designation ("100 g") was included.
- "Isolated" was replaced with "pure".
Reviewer 2 Report
This manuscript gives an overview of phytonutrients that are considered to be anti-nutrients. Overall, there is a lot of very good information in this manuscript and its well written. My main suggestions have to do with this being mentioned as a review. 1. If you want to keep this as a review, I would put the word Narrative Review in that than a critical review. There is also no methods section listed: Research question, aims, objectives, databases searched, search terms etc. 2. For each nutrient, you have incorporated many different aspects. I think putting subheadings into each would assist: Definition, Background, Digestion/Cooking, Safety, In vitro and preclinical studies (Animal), Clinical trials, Discussion, Conclusion 3. I know this is a long manuscript and you have done a conclusion at the end (which was the first time I saw your research question and purpose). I would section these out. I would do a limitations section, a summary, and then a small conclusion to finish the paper.
Author Response
- The title of the paper was changed to state that it is a "Narrative Review" rather than a critical review.
- The research question and additional subheadings were included for clarity.
- A "Limitations" section was added, along with a summary at the end.